# OpenReview forum: "MTSTRec: Multimodal Time-Aligned Shared Token Recommender"
_ICLR.cc/2025/Conference — ICLR 2025 Conference Withdrawn Submission_

### Official Review · Reviewer_eCz1 · 2024-10-28

**Soundness:** 2
**Presentation:** 3
**Contribution:** 1
**Rating:** 3
**Confidence:** 5

**Summary:**

MTSTRec is a multimodal recommendation model that aligns and fuses text, images, and price data to enhance e-commerce recommendations by capturing user preferences over time​

**Strengths:**

1.The paper is well-written and easy to follow.

2.It explores an interesting topic by incorporating the time factor into multimodal sequential recommendation.

3.This paper incorporates multiple modalities, providing a comprehensive approach.

**Weaknesses:**

1.The novelty of this paper is somewhat limited, as this line of research is well-established in the field of recommender systems. However, the authors do not discuss these existing approaches in the related work section nor include them as baseline comparisons [1,2,3].

2.The baseline approaches are insufficient. The authors should include more state-of-the-art multimodal recommendation methods, such as those in [4,5].

3.The motivation in the introduction is somewhat unclear. For instance, in the statement, "In contrast, late fusion methods process each modality separately, addressing modality-specific challenges but overlook the fact that at each time step, different modalities correspond to the same product," the rationale for why late fusion models overlook this aspect should be further elaborated. Since this issue is central to the paper's contribution, the authors should explain why late fusion models miss this detail.

4.The authors have neither provided the code nor promised to release it later, raising concerns about the reproducibility of the work.

[1] Campos, Pedro G., Fernando Díez, and Iván Cantador. "Time-aware recommender systems: a comprehensive survey and analysis of existing evaluation protocols." *User Modeling and User-Adapted Interaction* 24 (2014): 67-119.

[2] Zhang, Qi, et al. "Neural time-aware sequential recommendation by jointly modeling preference dynamics and explicit feature couplings." *IEEE Transactions on Neural Networks and Learning Systems* 33.10 (2021): 5125-5137.

[3] Jiang, Hao, et al. "What aspect do you like: Multi-scale time-aware user interest modeling for micro-video recommendation." *Proceedings of the 28th ACM International conference on Multimedia*. 2020.

[4] Zhou, Xin, and Zhiqi Shen. "A tale of two graphs: Freezing and denoising graph structures for multimodal recommendation." *Proceedings of the 31st ACM International Conference on Multimedia*. 2023.

[5] Zhong, Shanshan, et al. "Mirror Gradient: Towards Robust Multimodal Recommender Systems via Exploring Flat Local Minima." *Proceedings of the ACM on Web Conference 2024*. 2024.

**Questions:**

1. How do you substantiate the claim that "textual data alone is often sufficient for identifying what a product is, while images should capture product aesthetics and style, which are especially important on platforms offering the same product in various patterns or designs"? Could you provide some rationale behind this?

2. In the statement, "In contrast, late fusion methods process each modality separately, addressing modality-specific challenges but overlook the fact that at each time step, different modalities correspond to the same product," could you explain why these methods overlook this aspect?

3. How much additional computational cost does this model introduce compared to the original sequential recommendation models?

4. How do you represent the "time step" in your study? Without precise timestamps, it seems more like positional information rather than exact timing. In sequential recommendation, there's often a risk of time information leakage, such as inadvertently using future items to train current models. Have the authors accounted for this in their approach?

---

> ### Author Response · Authors · 2024-11-22
>
> Dear Reviewer,
> Thank you for taking the time to review our work and providing constructive feedback that will undoubtedly help us improve our research. Due to the limited time we had during the rebuttal period, we decided to withdraw our paper and spend more time improving it based on all reviewers’ comments. We thank you again for your time and valuable feedback.

---

### Official Review · Reviewer_2FA6 · 2024-10-30

**Soundness:** 3
**Presentation:** 3
**Contribution:** 3
**Rating:** 5
**Confidence:** 4

**Summary:**

This paper presents MTSTRec, an innovative Multimodal Time-aligned Shared Token Recommender designed to effectively integrate and transmit critical information across multiple modalities. This method enables an accurate fusion of multimodal features—including product IDs, images, text, and prices—while preserving the distinct contributions of each modality.

**Strengths:**

1.	The paper introduces a novel Time-aligned Shared Token (TST) module that uses shared tokens to capture cross-modal interactions at each time step in the sequence, ensuring time-consistent alignment and fusion of information from different modalities.
2.	Two new datasets for multimodal sequential recommendation tasks are developed and will be made available for future research.
3.	Experimental results validate the proposed model's effectiveness.

**Weaknesses:**

1.	Table 2 indicates that some modalities, such as Price, have minimal impact on model performance. It raises a concern regarding the selection of relevant modalities for the model.
2.	The multimodal fusion strategy is commonly employed in other multimodal tasks, limiting the perceived novelty.
3.	The paper contains some typographical and minor errors, such as an equation number error in line 699 on page 13.

**Questions:**

The experiments lack comparisons with some multimodal-specific recommendation models. Additionally, general recommendation models should incorporate multimodal information for fairer comparisons.

---

> ### Author Response · Authors · 2024-11-22
>
> Dear Reviewer,
> Thank you for taking the time to review our work and providing constructive feedback that will undoubtedly help us improve our research.
> Due to the limited time we had during the rebuttal period, we decided to withdraw our paper and spend more time improving it based on all reviewers’ comments. We thank you again for your time and valuable feedback.

---

### Official Review · Reviewer_hgRF · 2024-11-02

**Soundness:** 3
**Presentation:** 3
**Contribution:** 3
**Rating:** 3
**Confidence:** 4

**Summary:**

The paper proposes MTSTRec, a multimodal recommendation framework utilizing a transformer-based model with a Time-aligned Shared Token (TST) fusion module. By aligning data types—such as product IDs, text, images, and prices—along a temporal dimension, MTSTRec seeks to enhance cross-modal interactions in sequential recommendations. The authors present extensive experiments across multiple datasets, aiming to establish the model’s edge over existing multimodal systems. Overall, the paper highlights MTSTRec’s capacity to adapt and capture user preferences across various modalities.

**Strengths:**

-  Fusion Technique: The TST module is a commendable attempt at efficient, time-consistent multimodal feature fusion, capturing both modality-specific and cross-modal interactions.

-  Performance Gains: MTSTRec demonstrates strong performance across datasets, surpassing traditional and multimodal baselines, which speaks to its robustness in sequential recommendation tasks.
- Detailed Experiments: The authors provide comprehensive ablation studies and comparisons, shedding light on the role of individual features and the TST fusion mechanism, adding clarity to the model’s contributions.

**Weaknesses:**

- Limited Originality in Fusion Mechanism: The Time-aligned Shared Token (TST) fusion module appears to mirror existing multimodal fusion techniques, notably those used in large language models like Flamingo and other variants. This resemblance raises questions regarding the novelty of the fusion strategy.


- Narrow Scope of Evaluation: The experimental validation relies heavily on e-commerce datasets, which limits insights into the model’s generalizability. The adaptability of MTSTRec to other data-rich domains is not sufficiently demonstrated, leaving its versatility in question.

**Questions:**

1. What is MTSTRec’s computational efficiency compared to baseline models in real-time settings? Are there specific optimizations that could address any additional complexity introduced by the TST module?

2. Has there been any assessment of the model’s robustness to noisy or sparse data, which are frequently encountered in real-world recommendation systems?

---

> ### Author Response · Authors · 2024-11-22
>
> Dear Reviewer,
> Thank you for taking the time to review our work and providing constructive feedback that will undoubtedly help us improve our research. Due to the limited time we had during the rebuttal period, we decided to withdraw our paper and spend more time improving it based on all reviewers’ comments. We thank you again for your time and valuable feedback.

---

### Official Review · Reviewer_fp3d · 2024-11-05

**Soundness:** 3
**Presentation:** 3
**Contribution:** 2
**Rating:** 6
**Confidence:** 4

**Summary:**

The paper proposes MTSTRec, short for **M**ultimodal **T**ime-aligned **S**hared **T**oken **Rec**ommender, a novel multimodal recommender system for sequential recommendation. Unlike previous similar approaches, MTSTRec exploits multiple modalities to represent products in the catalog (such as the product image, textual description, and price); moreover, it employs a transformer-based architecture to provide cross-modality fusion that works also in time.

To begin with, the framework extracts the item's features, namely, the item ID, its visual style (by means of the Gram matrices), and the textual characteristics (obtained both from the product titles and descriptions through an LLM text encoder, and a prompt-text extractor that also leverages LLMs). Then, through a multimodal transformer, the framework learns distinct embeddings for each modality in the user's items sequence, while a self-attention encoder processes each embedding making them suitable for the subsequent TST module. Finally, the proposed TST module (inspired by the bottleneck mechanism) makes sure that multimodal embeddings for each item are fused while being aligned in time. The final loss function is built through a traditional binary cross-entropy loss.

The experiments largely demonstrate the efficacy of the MTSTRec proposed approach over a large set of similar baselines, on three multimodal and sequential recommendation datasets. Moreover, an ablation study on the influence of each modality shows how the ID one seems to provide the biggest contribution to the overall performance, followed by the text & prompt modality. Then, another ablation study on the fusion module proves that the introduced TST module is indeed beneficial in providing better recommendation results. Finally, by comparing different shared tokens configurations, the authors outline how the proposed one is again the one performing the best among all.

**Strengths:**

\+ The rationale of when to perform the fusion within a multimodal/sequential-based recommender system is quite interesting and under-debated in the literature

\+ The authors present the related work in detail and contextualize the proposed approach within it in an adequate manner

\+ I appreciated the different multimodal features applied, especially the style extractor and the prompt-text extractor ones

\+ The experimental settings is extensive, with several considered dimensions

\+ Code and datasets are released at review time

**Weaknesses:**

\- Results from Table 2 may underline some weaknesses of the proposed approach. While it is more understandable that the joint application of text and prompt could be crucial in delivering high-quality recommendations, it also appears that the contribution of the sole ID features is enough to obtain high-accuracy performance; this outcome would suggest that the multimodal contribution is not the most important one, rising some doubts regarding the contribution of the model, presented as a multimodal approach.

**Questions:**

The only question refers to the outlined weakness: what is the real contribution of multimodal features in this architecture, considering the ablation study conducted in Table 2? The authors may refer to this recent paper [\*], where it is debated whether pure multimodal-based recommender systems could outperform pure ID-based recommender systems. While the authors in [\*] do not come to an overall and final conclusion, they observe (among others) that modern pure multimodal approaches can outperform pure ID ones.

**References**

[\*] Zheng Yuan, Fajie Yuan, Yu Song, Youhua Li, Junchen Fu, Fei Yang, Yunzhu Pan, Yongxin Ni: Where to Go Next for Recommender Systems? ID- vs. Modality-based Recommender Models Revisited. SIGIR 2023: 2639-2649

**After the rebuttal**
I acknowledge the authors' decision to withdraw the paper.

---

> ### Author Response · Authors · 2024-11-22
>
> Dear Reviewer,
> Thank you for taking the time to review our work and providing constructive feedback that will undoubtedly help us improve our research. Due to the limited time we had during the rebuttal period, we decided to withdraw our paper and spend more time improving it based on all reviewers’ comments. We thank you again for your time and valuable feedback.

---

> > ### Comment · Reviewer_fp3d · 2024-11-22
> > **I acknowledge authors' decision**
> >
> > Dear Authors, thank you for your feedback, and good luck with the future directions of the paper! Indeed, I see potentially positive aspects that could be leveraged to improve the work and address other venues. If I were you, I'd especially focus, among others, on strengths #1 and #3 from my review. My 2 cents of course.

---

### Note · Authors · 2024-11-25

**Comment:**

Thanks for all reviewers taking the time to review our work and providing constructive feedback that will undoubtedly help us improve our research. Due to the limited time we had during the rebuttal period, we decided to withdraw our paper and spend more time improving it based on all reviewers’ comments. We thank you again for your time and valuable feedback.

**Withdrawal Confirmation:**

I have read and agree with the venue's withdrawal policy on behalf of myself and my co-authors.